# Trends in self-reported cost barriers to dental care in Ontario

**Mona Abdelrehim**[1]*, **Vahid Ravaghi**[1,2], **Carlos Quiñonez**[1,3], **Sonica Singhal**[1,4]

**1** Faculty of Dentistry, University of Toronto, Toronto, Canada, **2** School of Dentistry, University of Birmingham, Birmingham, United Kingdom, **3** Schulich School of Medicine & Dentistry, Western University, London, Canada, **4** Public Health Ontario, Toronto, Canada

* mona.abdelrehim@mail.utoronto.ca

## Abstract

### Background

The affordability of dental care continues to receive attention in Canada. Since most dental care is privately financed, the use of dental care is largely influenced by insurance coverage and the ability to pay-out-of pocket.

### Objectives

i) to explore trends in self-reported cost barriers to dental care in Ontario; ii) to assess trends in the socio-demographic characteristics of Ontarians reporting cost barriers to dental care; and iii) to identify the trend in what attributes predicts reporting cost barriers to dental care in Ontario.

### Methods

A secondary data analysis of five cycles (2003, 2005, 2009–10, 2013–14 and 2017–18) of the Canadian Community Health Survey (CCHS) was undertaken. The CCHS is a cross-sectional survey that collects information related to health status, health care utilization, and health determinants for the Canadian population. Univariate and bivariate analyses were conducted to determine the characteristics of Ontarians who reported cost barriers to dental care. Poisson regression was used to calculate unadjusted and adjusted prevalence ratios to determine the predictors of reporting a cost barrier to dental care.

### Results

In 2014, 34% of Ontarians avoided visiting a dental professional in the past three years due to cost, up from 22% in 2003. Having no insurance was the strongest predictor for reporting cost barriers to dental care, followed by being 20–39 years of age and having a lower income.

### Conclusion

Self-reported cost barriers to dental care have generally increased in Ontario but more so for those with no insurance, low income, and aged 20–39 years.

**Data Availability Statement:** The Public Use Microdata Files (PUMF) for the Canadian Community Health Survey (CCHS) data were accessed online using the Survey Documentation

and Analysis (SDA) online tool available through the University of Toronto library at the Computing in the Humanities and Social Sciences (CHASS) portal. Requests and further information on accessing the dataset can be obtained here: https://mdl.library.utoronto.ca/research/help.

**Funding:** The author(s) received no specific funding for this work.

**Competing interests:** The authors have declared that no competing interests exist.

## 1. Introduction

In Canada, physician and hospital-based services are publicly funded under the Canadian health care system [Medicare] [1]. In contrast, Canadians are primarily responsible for privately financing their dental care except for surgical-dental services delivered in hospitals which are publicly funded [2, 3]. In 2018, approximately 94% of dental care expenditure was privately financed, and the remaining 6% was publicly financed (through federal, provincial/ territorial and municipal governments) targeting vulnerable and socially marginalized groups. Approximately 61% of the privately financed dental care was paid through insurance (employer-sponsored coverage or self-insured), and the remaining 39% was through out-of-pocket payments. Important to note that this 39% of payment is not only coming from those who have no form of private insurance coverage but also from the insured individuals as despite coverage, people pay some proportion of their dental care bill as a co-payment (generally 20% to 50%) from out-of-pocket [4]. Thus, with a large portion of Canadians financing their dental care, cost becomes the predominant factor limiting access to care [4, 5].

Affordability is one of the five dimensions of access to health services [6]. Previous studies have examined the affordability of dental care by asking individuals whether they considered the cost of dental care a burden and whether they avoided visiting a dentist or even declined treatment due to cost [4, 7, 8]. Moreover, national surveys such as the Canadian Community Health Survey (CCHS) since 2003 have been asking participants whether they had previously avoided visiting a dental professional due to cost [9–13]. The cycle 1 of the Canadian Health Measures Survey (CHMS) [2007–09] also asked participants whether they had declined a recommended dental treatment due to cost in the past [14]. Sadly, significant number of Canadians do report cost barriers to dental care, where cost is the second most prevalent reason for not visiting a dentist, after lack of dental insurance [15].

It is important to emphasize that the unaffordability of dental care has been a significant concern in other countries as well. In the United States, for example, cost has been reported as one of the main barriers to dental care. Although cost barriers for children fell from 2005 to 2019, there was an increasing trend among adults and seniors [16]. Similarly, Australians reported cost as the primary reason for not visiting a dentist [17]. Studies have shown that the proportion of Australians who avoided or delayed visiting a dentist due to cost increased from 27% in 1994 to 34% in 2008 and to 39% in 2017–18 [18–20]. In the United Kingdom, while clinically necessary dental treatment is covered under the National Health Service (NHS), there are concerns about the affordability of dental care. Results from the 2009 Adult Dental Health Survey revealed that almost one-fifth of respondents delayed dental treatment due to cost, and around 25% claimed that the cost of treatment had affected the type of treatment they had chosen in the past [21]. Additionally, 43% of survey participants in 2010 avoided visiting a dentist due to cost [22].

Income and insurance status are known to be the dominant predictors in reporting cost barriers to dental care [23–25]. Findings from previous studies confirm that low-income Canadians are more likely to report financial barriers to dental care than their high-income counterparts [4, 7, 23, 26]. Locker et al. found that 28% of low-income participants considered the cost of dental care a burden, therefore avoiding visiting a dentist and declining the recommended treatment due to cost compared to 5% of the high-income group [7]. It is also important to note that the challenge in accessing dental care in Canada is not only related to income but also to the type of jobs, as precarious, part-time or contracted jobs, generally do not qualify for employment- based insurance [27, 28].

Previous studies revealed that inequality in oral health is common in Canada, with higher-income groups more likely to receive care than lower-income groups [29–31]. Further, the

literature has shown that financial barriers to dental care are associated with poor oral health and routine dental attendance is associated with better oral health [23, 32]. Thus, inequality in oral health can be attributed at least, in part, to inequity in access to and utilization of care. In other words, Canada observes the "inverse care law" in dental care, as people with highest needs utilize dental care the least and vice versa [33, 34].

Through CCHS 2018 data, we know that 22.4% of Canadians avoided visiting a dentist due to cost [35]; however, we do not know the trends in cost barriers of dental visits over the last two decades. Importantly, we do not know what socio-economic and demographic factors have consistently attributed to these cost barriers and what have changed over time. Understanding these trends and their attributable factors would support policymakers to have a targeted approach to address such barriers, notably with the recent growing policy and society's interest in expanding oral healthcare coverage in Canada. Recently, Prime Minister Justin Trudeau announced plans to establish a national dental care program for low-income Canadians. [36–38]. Moreover, a recent study showed that inequalities in dental care are narrowing in most Canadian provinces except for Ontario [39]. It is for this reason that our study uses data from Ontario, Canada's most populated province, and our objectives are i) to explore trends in self-reported cost barriers to dental care in Ontario, ii) to assess trends in the socio-demographic characteristics of Ontarians reporting cost barriers to dental care, and iii) to identify the trend in what attributes predicts reporting cost barriers to dental care in Ontario.

## 2. Methodology

### 2.1 Study design and sample

Our study is a secondary data analysis of five cycles of a repeated cross-sectional national survey, the Canadian Community Health Survey (CCHS) (cycles: 2003, 2005, 2009–10, 2013–14 and 2017–18]. The Public Use Microdata Files (PUMF) for the five CCHS data cycles were accessed online using the Survey Documentation and Analysis (SDA) online tool available through the University of Toronto Library at the Computing in the Humanities and Social Sciences (CHASS) portal. No ethics review was sought for the study, as this was a secondary data analysis of anonymized data that contain no personal identifiers, nor was it linked to any other data source [40].

In order for estimates produced from the CCHS survey data to be representative of the covered population and not merely of the sample itself, survey weights were applied during the data analysis for this study. A survey weight was assigned to each respondent included in the final sample. This weight corresponds to the number of people in the whole population that the respondent represents.

### 2.2 Data collection

The CCHS is a national population based cross-sectional survey representing approximately 97% of the Canadian population. The survey collects information related to health status, health care utilization and health determinants for the Canadian population at the regional and provincial levels. It targets individuals, aged 12 years or older, living in private dwellings in the ten provinces and the three territories. Excluded from the survey's coverage are people living on reserves and other Indigenous settlements in the provinces, full-time members of the Canadian Forces and the institutionalized population. By the 2017–18 cycle, youth aged 12 to 17 living in foster homes were also excluded from the survey [9–13]. Data is collected over the telephone or in-person, using computer-assisted personal interviewing (CAPI) or computer-assisted telephone interviewing (CATI) techniques. The combined response rate for the 2003 cycle was 80.7% and declined to 58.8% by the 2017–18 cycle [9–13].

## 2.3 Variables of interest

The outcome variable in this study is avoiding visiting a dental professional due to cost. In 2003, 2005, 2009–10 and 2013–14 cycles, the question asked was, "*What are the reasons that you have not been to a dentist in the past 3 years?*". There were 15 reasons to select from, including "avoiding due to cost," and respondents could mark all that apply. However, in the 2017–18 cycle the question was "In the past 12 months, have you avoided going to a dental professional because of the cost of dental care?" to which respondents answered "yes" or "no." A similar question has been used by the Canadian Health Measures Survey in its first cycle (2007–09).

Covariates selected for this study were based on Andersen's model of health care utilization [41] and existing scientific literature that identifies a number of socio-economic and demographic attributes that create cost barriers to access to care [4, 7, 23–25, 27, 42]. A number of questions vary by cycles and provinces/territories; therefore, it was crucial to review and compare questions over the CCHS cycles to ensure their consistency, and we found that the most consistent data available for the CCHS cycles was for the province of Ontario, and the most complete data was for the following five cycles: 1) 2003; 2) 2005; 3) 2009–10; 4) 2013–14, and 5) 2017–18. Fig 1 illustrates conceptualizing and grouping of the selected variables in this study according to the Anderson model of health care utilization.

Predisposing characteristics include demographic characteristics, one's social structure and those variables believed to predispose individuals to report cost barriers to dental care. This study includes age, sex, marital status and education as predisposing characteristics. We categorized age into five groups: "12–19," "20–39," "40–59," "60–79" and "80 and older". Marital status was recoded and categorized into three groups "married/common law," "widowed/divorced/separated," and "single." Education is indicated by the highest level of education of any member of the household and was dichotomized into "≤ secondary school graduation" and "> secondary school graduation" to maintain consistency with other literature on this topic [23, 25].

In terms of enabling variables, these variables include both community and personal resources and are required to report cost barriers to dental care, including income, dental insurance and employment status in this study. Household income is a proxy for socio-economic status and it reflects an individual's ability to afford dental care. Income in the 2003 cycle was reported as "income adequacy," a derived variable, and was classified into five categories based on the respondent's total household income and the number of people living in the household (Appendix 1 describes how each income category was derived). For 2005, 2009–10, 2013–14 and 2017–18 cycles, household income was reported as deciles at the provincial level based on the adjusted household income ratio to a standard low-income cut-off accounting for household and community size. We collapsed the ten deciles into five quintiles for

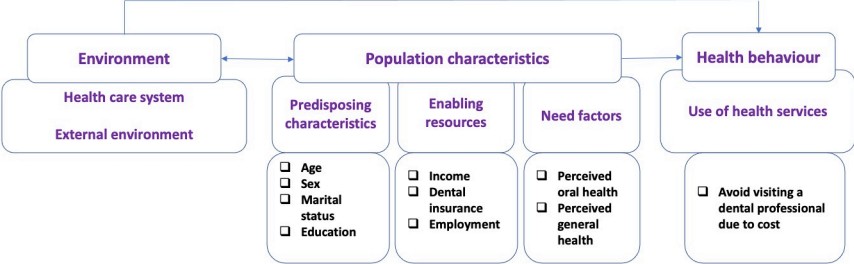

**Fig 1. Grouping of the selected variables in this study according to the Anderson model of health care utilization.**

analysis. The lowest quintile represents the lowest 20% of adjusted ratios, while the highest quintile represents the highest 20% of ratios. Since it is well known that dental insurance somehow offsets the cost of dental care, information regarding the availability of dental insurance "yes" or "no" was available for all five cycles and derived from the following question: "*Do you have insurance that covers all or part of your dental expenses*?". For all cycles except the 2003 cycle, there was a question asking about the type of dental insurance "employment-based," "government-based" or "private" insurance. The employment status variable for this study was derived from two questions: "Have you worked at a job or business at any time in the past 12 months? and "On average, how many hours do you usually work per week at your job(s)? The variable was grouped into three categories: "Full-time employed," "part-time employed," and "unemployed." Full-time was classified as working more than 30 hours per week, and part-time as less than 30 hours per week.

Lastly, needs factors in this study include how individuals view their health, in other words, self-perceived needs. In 2003, 2013–14 and 2017–18 cycles, participants were asked, "*In general, would you say the health of your mouth is. . .?*" using a five-point scale from poor to excellent. In our study, we grouped "good to excellent" and "fair to poor." Similarly, respondents were asked to rate their general health using the following categories: excellent, very good, good, fair, and poor. These categories were then dichotomized into "good to excellent" and "fair to poor," which were used in the analyses.

## 2.4 Statistical analysis

The CCHS data was exported to a Microsoft Excel (© Microsoft 365 for Mac) worksheet and then imported into Stata v.17 software (© StataCorp: Release 17) for statistical analysis [43]. We started with univariate and bivariate analyses to examine the sample characteristics and determine the characteristics of Ontarians, who reported cost barriers to dental care. Then Poisson regression was conducted to calculate unadjusted and adjusted prevalence ratios to determine the predictors of reporting a cost barrier to dental care. We used the prevalence ratio to avoid "overestimation" of the association by odds ratio [44–46]. The unadjusted prevalence ratio, 95% confidence interval and p-value were assessed.

The most important independent variables were selected to enter into a single regression model based on previous literature and the significance level was set at $p < 0.1$, which is acceptable in social-ecological studies. Moreover, collinearity amongst and between the variables was assessed, and only variables with variance inflation factor VIF<3 were entered into the model. The VIF is a measure that quantifies the severity of multicollinearity. It is the inverse of tolerance and represents the extent to which variances are inflated or increased due to collinearity [47]. The adjusted prevalence ratio, 95% confidence interval and p-value were recorded for the variables in the regression model.

## 3. Results

### 3.1 Surveys sample characteristics

The baseline characteristics of Ontarians in the five cycles of the CCHS based on predisposing, enabling and needs factors are presented in Table 1. Overall, 203,112 Ontarians participated in the five cycles of CCHS.

### 3.2 Trends in cost barriers to dental care in Ontario

Due to variation in the time frame of the question addressing cost barriers to dental care over the five cycles, Fig 2 has two components. Part (a) shows trends in cost barriers to dental care

**Table 1. Baseline characteristics of Ontarians in the five cycles of the Canadian Community Health Survey (CCHS).**

| Characteristics | Weighted (%) (95% CI) | | | | |
|---|---|---|---|---|---|
| | **2003** | **2005** | **2009–10** | **2013–14** | **2017–18** |
| **Predisposing factors** | | | | | |
| **Age** | | | | | |
| 12–19 | 12.6 (12.2, 13.1) | 12.6 (12.1, 13.0) | 12.0 (11.6, 12.5) | 11.0 (10.5, 11.5) | 10.5 (9.9, 11.1) |
| 20–39 | 33.7 (33.0, 34.5) | 32.7 (32.0, 33.4) | 30.9 (30.1, 31.7) | 31.0 (30.2, 31.9) | 31.2 (30.3, 32.2) |
| 40–59 | 34.4 (33.7, 35.1) | 34.7 (34.0, 35.5) | 35.5 (34.6, 36.4) | 33.9 (33.0, 34.9) | 31.8 (30.9, 32.8) |
| 60–79 | 16.4 (16.0, 16.9) | 16.9 (16.5, 17.4) | 17.9 (17.4, 18.4) | 20.3 (19.7, 20.9) | 22.3 (21.6, 23.0) |
| >80 | 2.8 (2.6, 3.0) | 3.1 (2.9, 3.3) | 3.7 (3.4, 3.9) | 3.8 (3.6, 4.0) | 4.2 (4.0, 4.5) |
| **Sex** | | | | | |
| Male | 49.1 (48.4, 49.9) | 49.2 (48.4, 49.9) | 49.0 (48.2, 49.9) | 48.9 (48.0, 49.8) | 48.9 (47.9, 49.9) |
| Female | 50.9 (50.2, 51.6) | 50.8 (50.1, 51.6) | 51.0 (50.1, 51.8) | 51.1 (50.2, 52.0) | 51.1 (50.2, 52.1) |
| **Marital status** | | | | | |
| Married/ Common law | 58.8 (58.1, 59.5) | 59.1 (58.3, 59.8) | 58.4 (57.6, 59.2) | 57.4 (56.5, 58.8) | 57.3 (56.4, 58.3) |
| Widowed/divorced/ separated | 11.0 (10.6, 11.4) | 10.7 (10.3, 11.1) | 11.8 (11.3, 12.3) | 11.9 (11.4, 12.5) | 11.7 (11.2, 12.2) |
| Single | 30.2 (29.5, 30.9) | 30.2 (29.6, 30.9) | 29.8 (29.2, 30.6) | 30.7 (29.9, 31.5) | 31.0 (30.0, 31.9) |
| **Highest level of education of household** | | | | | |
| ≤ secondary school graduation | 20.5 (20.0, 21.1) | 17.5 (17.0, 18.0) | 17.5 (16.9, 18.1) | 18.1 (17.4, 18.7) | 18.0 (17.3, 18.7) |
| > secondary school graduation | 79.5 (78.9, 80.0) | 82.5 (82.0, 83.0) | 82.5 (81.9, 83.2) | 82.0 (81.3, 82.6) | 82.0 (81.3, 82.7) |
| **Enabling factors** | | | | | |
| **Household income group** | | | | | |
| First (lowest) | 2.4 (2.2, 2.7) | 18.4 (17.8, 19.0) | 19.9 (19.2, 20.7) | 21.0 (20.2, 21.8) | 20.1 (19.3, 20.9) |
| Second | 4.9 (4.6, 5.3) | 18.1 (17.5, 18.7) | 19.9 (19.1, 20.7) | 19.5 (18.8, 20.3) | 19.5 (18.7, 20.3) |
| Third | 17.0 (16.5, 17.6) | 19.8 (19.1, 20.4) | 19.2 (18.5, 19.9) | 19.8 (19.1, 20.6) | 19.8 (19.1, 20.6) |
| Fourth | 32.8 (32.1, 33.6) | 20.7 (20.1, 21.3) | 20.2 (19.5, 21.0) | 19.6 (18.9, 20.3) | 19.7 (18.9, 20.5) |
| Fifth (highest) | 42.8 (42.0, 43.6) | 23.1 (22.4, 23.7) | 20.8 (20.1, 21.6) | 20.1 (19.4, 20.8) | 20.9 (20.1, 21.7) |
| **Dental insurance** | | | | | |
| Yes | 67.6 (66.8, 68.4) | 67.7 (67.0, 68.4) | 66.1 (65.3, 67.0) | 66.9 (66.0, 67.8) | 67.2 (66.2, 68.1) |
| No insurance | 32.4 (31.6, 33.2) | 32.3 (31.6, 33.0) | 33.9 (33.1, 34.7) | 33.10 (32.2, 34.0) | 32.8 (31.9, 33.8) |
| **Type of dental insurance** | | | | | |
| Employment-based insurance | N/A* | 57.5 (56.8, 58.2) | 55.7 (54.8, 56.5) | 55.4 (54.5, 56.4) | 57.8 (56.8, 58.8) |
| Government insurance | | 5.3 (5.0, 5.6) | 5.7 (5.4, 6.1) | 6.1 (5.7, 6.6) | 4.3 (3.9, 4.7) |
| Private insurance | | 4.5 (4.2, 4.9) | 4.3 (4.0, 4.7) | 4.9 (4.6, 5.3) | 4.0 (3.6, 4.4) |
| No insurance | | 32.7 (32.0, 33.4) | 34.3 (33.5, 35.1) | 33.5 (32.64, 34.4) | 33.9 (32.9, 34.9) |
| **Employment status** | | | | | |
| Full-time employed | 63.7 (62.9, 64.4) | 63.5 (62.7, 64.3) | 61.0 (60.1, 61.9) | 60.5 (59.5, 61.5) | 60.7 (59.7, 61.8) |
| Part-time employed | 14.5 (13.9, 15.0) | 12.6 (12.0, 13.1) | 13.1 (12.5, 13.7) | 12.5 (11.8, 13.2) | 11.8 (11.1, 12.6) |
| Unemployed | 21.9 (21.3, 22.5) | 23.9 (23.3, 24.6) | 25.9 (25.1, 26.7) | 27.1 (26.2, 27.9) | 27.5 (26.6, 28.4) |
| **Needs factors** | | | | | |
| **Perceived oral health** | | | | | |
| Good to Excellent | 85.2 (84.7, 85.8) | N/A* | N/A* | 85.2 (84.4, 85.8) | 89.7 (89.1, 90.2) |
| Fair to Poor | 14.8 (14.2, 15.3) | | | 14.8 (14.2, 15.6) | 10.3 (9.8, 10.9) |
| **Perceived general health** | | | | | |
| Good to Excellent | 88.2 (87.7, 88.6) | 89.0 (88.5, 89.4) | 88.1 (87.6, 88.7) | 88.3 (87.7, 88.8) | 88.9 (88.3, 89.4) |
| Fair to Poor | 11.8 (11.4, 12.3) | 11.0 (10.6, 11.50) | 11.9 (11.4, 12.4) | 11.7 (11.2, 12.3) | 11.1 (10.6, 11.7) |

*Information on this data variable not collected in the specific survey cycle

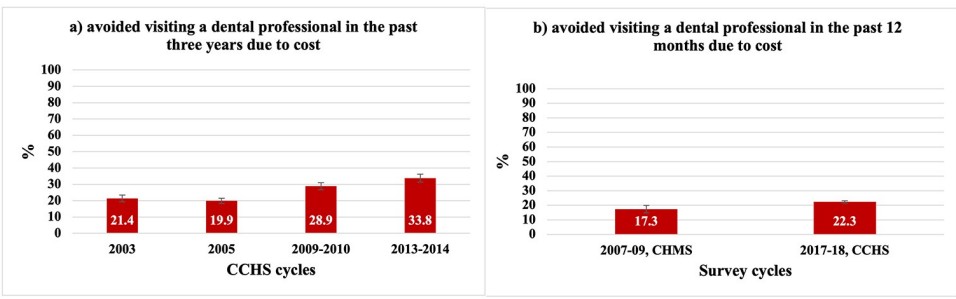

**Fig 2. Trends in self-reported cost barriers to dental care.** a) avoided visiting a dental professional in the past three years due to cost and b) avoided visiting a dental professional in the past 12 months due to cost.

in the past three years prior to the survey and includes the first four CCHS cycles, and part (b) shows trends in cost barriers to dental care in the past 12 months prior to the survey and includes the 2017–18 CCHS cycle, which is compared to the 2007–09 CHMS cycle 1 data.

From 2003 to 2013–14, people who avoided visiting a dental professional in the last three years due to cost increased from 21.4% (95% CI: 19.9, 24.1) to 33.8% (95% CI: 31.4, 36.4). Similarly, from 2007–09 to 2017–18, people who avoided visiting a dental professional in the past 12 months increased from 17.3% of all Canadians (95% CI: 14.9, 20.0)to 22.3% (95% CI: 21.5, 23.2) for Ontarians.

## 3.3 Characteristics of Ontarians who experienced cost barriers to dental care

Table 2 outlines the characteristics of Ontarians who experienced cost barriers to dental care from 2003 to 2017–18.

**3.3.1 Predisposing factors.** Table 2, shows that among all groups, people aged 20 to 39 have consistently been the highest proportion reporting cost barriers to dental care followed by those aged 40–59 and the least was for those aged 80 years and more. Irrespective of the age, the proportion of those facing financial barriers increased in all age groups over the years. Only for 12–19 years old, the proportions reduced from 2007–09 to 2017–18: however, the change was not significant.

In terms of education, from 2003 to 2013–14; people with higher than secondary school education reported more cost barriers than their counterparts: however, the trend reversed in 2917–18. As per the CHMS cycle 1, there was no difference among Canadians based on their education level. Again, overall the cost barriers to access oral health care increase among all, irrespective of the education level.

**3.3.2 Enabling factors.** Table 2 shows that over the five CCHS five cycles, Ontarians from the lowest household income group experienced the most cost barriers to dental care than other income groups. That said, the 2003 cycle could not be included in the trends, because in 2003, household income was represented by income adequacy while in 2005, 2009–10 and 2013–14 cycles were represented by income deciles. From 2005 to 2013–14, the proportion of Ontarians who reported financial barriers to dental care increased among all income groups. Importantly, the second and third income groups showed the largest rise. Unfortunately, we do not have data from the CHMS 2007–09 to compare to.

Over the five cycles, Ontarians with no dental insurance experienced significantly more cost barriers to dental care than those with dental insurance, and the proportion of those who faced barriers increased over the years.

**Table 2. Avoiding visiting a dental professional due to cost according to socio-demographic factors in the five cycles of the Canadian Community Health Survey (CCHS) and the 2007–09 cycle of the Canadian Health Measures Survey (CHMS).**

| Characteristics | Weighted (%) (95% CI) | | | | | |
|---|---|---|---|---|---|---|
| | CCHS | | | | | CHMS |
| | **2003** | **2005** | **2009–10** | **2013–14** | **2017–18** | **2007–09** |
| **Predisposing factors** | | | | | | |
| **Age** | | | | | | |
| 12–19 | 20.1 (13.8, 28.30) | 18.7 (13.2, 26.0) | 30.0 (20.6, 41.3) | 27.3 (16.2, 42.1) | 7.4 (5.8, 9.3) | 9.5 (7.2, 12.4) |
| 20–39 | 34.9 (30.7, 39.4) | 35.1 (31.7, 38.6) | 42.5 (38.5, 46.6) | 44.6 (39.7, 49.7) | 29.5 (27.7, 31.3) | 23.7 (19.1, 29.0) |
| 40–59 | 24.7 (20.7, 29.2) | 20.9 (17.7, 24.6) | 34.5 (29.3, 40.0) | 40.6 (35.4, 46.0) | 21.2 (19.7, 22.7) | 17.5 (14.4, 21.2) |
| 60–79 | 8.6 (6.6, 11.1) | 6.6 (5.3, 8.3) | 13.2 (11.2, 15.6) | 21.1 (18.2, 24.3) | 21.7 (20.2, 23.2) | 13.2 (10.7, 16.2) |
| >80 | 2.4 (1.2, 5.0) | 5.4 (2.9, 9.9) | 4.4 (2.7, 7.4) | 8.1 (5.2, 12.4) | 15.5 (13.1, 18.1) | N/A* |
| **Sex** | | | | | | |
| Male | 21.0 (18.2, 24.1) | 18.5 (16.5, 20.8) | 27.1 (24.0, 30.5) | 31.8 (28.5, 35.3) | 20.4 (19.2, 21.7) | 15.5 (12.4, 19.1) |
| Female | 23.1 (20.3, 26.8) | 21.6 (19.3, 24.1) | 30.9 (27.8, 34.2) | 36.6 (33.1, 40.3) | 24.1 (22.9, 25.3) | 19.2 (16.1, 22.7) |
| **Marital status** | | | | | | |
| Married/ Common law | 20.3 (17.7, 23.3) | 18.4 (16.5, 20.6) | 27.3 (24.2, 30.7) | 31.8 (28.3, 35.6) | 20.5 (19.4, 21.7) | N/A* |
| Widowed/divorced/ separated | 17.4 (14.0, 21.4) | 14.7 (11.6, 18.3) | 23.9 (20.2, 28.1) | 30.7 (26.8, 35.0) | 29.3 (27.38, 31.3) | |
| Single | 30.2 (25.5, 35.3) | 28.0 (24.4, 31.9) | 37.2 (32.7, 41.9) | 40.4 (35.5, 45.4) | 22.9 (21.3, 24.7) | |
| **Highest level of education of household** | | | | | | |
| ≤ secondary school graduation | 16.9 (14.1, 20.3) | 11.5 (9.7, 13.7) | 22.5 (18.3, 27.3) | 30.4 (38.0, 46.9) | 27.7 (25.9, 29.6) | 15.2 (12.2, 18.7) |
| > secondary school graduation | 23.9 (21.3, 26.8) | 22.9 (20.7, 25.2) | 32.1 (29.2, 35.1) | 35.2 (32.0, 38.6) | 21.1 (20.1, 22.1) | 17.9 (14.9, 21.3) |
| **Enabling factors** | | | | | | |
| **Household income group** | | | | | | |
| First (lowest) | 35.3 (22.9, 50.1) | 25.7 (22.6, 29.0) | 35.2 (31.2, 39.4) | 37.7 (33.4, 42.3) | 36.7 (34.5, 38.9) | N/A* |
| Second | 21.3 (15.7, 28.2) | 18.7 (15.3, 22.6) | 32.6 (27.3, 38.4) | 32.7 (28.2, 37.6) | 29.5 (27.5, 31.7) | |
| Third | 24.5 (20.3, 29.2) | 21.26 (17.5, 25.6) | 25.8 (19.4, 33.4) | 34.7 (29.0, 40.8) | 20.6 (18.8, 22.5) | |
| Fourth | 21.8 (18.1, 25.9) | 17.4 (13.5, 22.2) | 19.9 (14.0, 27.4) | 30.8 (24.4, 38.0) | 15.6 (13.8, 17.6) | |
| Fifth (highest) | 18.9 (14.8, 23.8) | 11.28 (8.1, 15.5) | 16.4 (11.7, 22.4) | 18.8 (13.4, 25.7) | 10.3 (9.0, 11.7) | |
| **Dental insurance** | | | | | | |
| Yes | 11.0 (8.6, 14.2) | 11.3 (9.4, 13.4) | 14.0 (11.6, 16.9) | 18.4 (15.1, 22.1) | 12.5 (11.7, 13.4) | N/A* |
| No insurance | 29.8 (26.9, 32.9) | 26.0 (23.7, 28.3) | 37.5 (34.4, 40.7) | 43.5 (40.1, 46.9) | 43.0 (41.2, 44.7) | |
| **Type of dental insurance** | | | | | | |
| Employment-based insurance | N/A* | 10.6 (8.7, 12.8) | 13.9 (11.16, 17.3) | 19.9 (15.9, 24.6) | **11.5 (10.7, 12.4) | **8.6 (7.3, 10.1) |
| Government insurance | | 10.8 (7.0, 16.3) | 11.4 (7.6, 16.7) | 17.1 (11.2, 25.2) | 24.4 (20.0, 29.4) | 8.9 (4.4, 17.1) |
| Private insurance | | 16.5 (8.2, 30.7) | 19.7 (9.3, 36.9) | 8.0 (3.2, 18.8) | | |
| No insurance | | 26.0 (23.7, 28.3) | 37.5 (34.4, 40.7) | 43.5 (40.1, 46.9) | 43.0 (41.2, 44.7) | 35.9 (30.4, 41.9) |
| **Employment status** | | | | | | |
| Full-time employed | 26.9 (23.7, 30.5) | 24.8 (22.2, 27.6) | 36.6 (32.3, 41.2) | 39.9 (35.6, 44.3) | 21.2 (20.0, 22.5) | 16.2 (13.0, 20.0) |
| Part-time employed | 38.2 (30.3, 46.9) | 25.6 (19.5, 32.8) | 40.3 (32.2, 48.9) | 43.4 (33.5, 53.8) | 28.5 (25.4, 31.9) | 20.0 (13.2, 29.0) |
| Unemployed | 16.3 (13.4, 19.7) | 14.0 (11.8, 16.5) | 24.29 (21.2, 27.6) | 33.2 (29.3, 37.4) | 25.4 (23.8, 27.2) | 20.7 (17.3, 24.5) |
| **Needs factors** | | | | | | |
| **Perceived oral health** | | | | | | |
| Good to Excellent | 15.0 (13.1, 17.1) | N/A* | N/A* | 27.6 (24.8, 30.6) | 19.0 (18.1, 19.9) | 13.6 (11.2, 16.3) |
| Fair to Poor | 40.3 (35.4, 45.4) | | | 49.4 (44.6, 54.2) | 50.5 (47.6, 53.4) | 37.9 (30.6, 45.7) |
| **Perceived general health** | | | | | | |
| Good to Excellent | 22.6 (20.3, 25.0) | 21.0 (19.23, 23.0) | 30.0 (27.5, 32.7) | 33.8 (31.0, 36.7) | 20.5 (19.6, 21.4) | N/A* |
| Fair to Poor | 19.5 (15.7, 24.0) | 14.95 (12.4, 18.0) | 23.9 (19.8, 28.5) | 34.1 (28.9, 39.6) | 38.6 (36.0, 41.3) | |

*Information on this data variable not collected in the specific survey cycle

**Employment-based insurance plus private insurance

Among those with insurance, an increasing trend in financial barriers to dental care was experienced by individuals with government-based or employment-based insurance: however, barriers seem to have reduced for those with private insurance.

From 2003 to 2013–14, Ontarians who worked part-time experienced more cost barriers to dental care than those who worked full-time or were unemployed' however, an increasing trend in the proportion of those avoided visiting a dental professional due to cost was observed among all full-time, part-time, or not-employed.

**3.3.3 Need factors.** Assessing by need, Ontarians who perceived their oral health as "fair to poor" experienced more financial barriers to dental care than those with "good to excellent" oral health. In terms of trends, irrespective of oral health, the proportions who perceived barriers to access to care increased from 2003 to 2013–14 and from 2007–09 to 2017–18.

## 3.4 Determining the strongest predictors of avoiding visiting a dental professional due to cost

Tables 3 and 4 show the adjusted prevalence ratio of individuals who avoided visiting a dental professional due to cost in the last three years and 12 months, respectively. After controlling for all other factors in the regression model, dental insurance, age and income were the three strongest predictors to avoid visiting a dental professional in the last three years due to cost across all four survey cycles: 2003; 2005; 2009–10; and 2013–14. The attributes remained consistent for the 2017–18 cycle.

Results of the adjusted prevalence ratios for the three strongest predictors to avoid visiting a dental professional due to cost in the last three years are graphically represented in Fig 3.

**Table 3. Adjusted prevalence ratio for avoiding visiting a dental professional in the past three years due to cost, from 2003-2013-14.**

| Independent variables | Canadian Community Health Survey Cycles | | | | | | | |
|---|---|---|---|---|---|---|---|---|
| | **2003** | | **2005** | | **2009–10** | | **20013–14** | |
| | **Adjusted Prevalence ratio (95% CI)** | **_P_-value** | **Adjusted Prevalence ratio (95% CI)** | **_P_-value** | **Adjusted Prevalence ratio (95% CI)** | **_P_-value** | **Adjusted Prevalence ratio (95% CI)** | **_P_-value** |
| **Age (12–19 as a reference group)** | | | | | | | | |
| 20–39 | 1.6 (1.0, 2.6) | 0.075 | 2.1 (1.2, 3.7) | 0.009 | 1.8 (1.0, 3.2) | 0.061 | 2.1 (1.2, 3.5) | 0.010 |
| 40–59 | 1.2 (0.7, 2.0) | 0.456 | 1.5 (0.8, 2.6) | 0.203 | 1.5 (0.8, 2.8) | 0.178 | 1.7 (1.0, 2.9) | 0.072 |
| 60–79 | 0.3 (0.2, 0.6) | <0.001 | 0.6 (0.3, 1.0) | 0.063 | 0.6 (0.3, 1.1) | 0.076 | 1.0 (0.6, 1.7) | 0.950 |
| >80 | N/A | N/A | N/A | N/A | N/A | N/A | N/A | N/A |
| **Sex (male as a reference group)** | | | | | | | | |
| Female | 1.2 (1.0, 1.5) | 0.100 | 1.4 (1.1, 1.6) | 0.001 | 1.2 (1.0, 1.5) | 0.037 | 1.2 (1.0, 1.4) | 0.020 |
| **Household Income Group (fifth (highest) as a reference group)** | | | | | | | | |
| First (lowest) | 1.4 (0.7, 2.7) | 0.305 | 2.3 (1.6, 3.3) | <0.001 | 1.9 (1.3, 2.7) | 0.001 | 1.8 (2.0, 2.6) | 0.004 |
| Second | 1.2 (0.7, 2.0) | 0.475 | 1.6 (1.1, 2.4) | 0.017 | 1.7 (1.2, 2.6) | 0.007 | 1.5 (1.0, 2.2) | 0.057 |
| Third | 1.2 (0.9, 1.8) | 0.239 | 1.7 (1.2, 2.5) | 0.006 | 1.6 (1.1, 2.4) | 0.027 | 1.9 (1.3, 2.9) | 0.001 |
| Fourth | 1.3 (0.9, 1.7) | 0.131 | 1.7 (1.1, 2.5) | 0.011 | 1.2 (0.7, 1.9) | 0.462 | 1.6 (1.1, 2.4) | 0.028 |
| **Dental Insurance (yes as a reference group)** | | | | | | | | |
| No insurance | 3.1 (2.2, 4.2) | <0.001 | 2.6 (2.1, 3.3) | <0.001 | 2.8 (2.2, 3.6) | <0.001 | 2.7 (2.2, 3.4) | <0.001 |
| **Employment Status (full-time employed as a reference group)** | | | | | | | | |
| Part-time employed | 1.1 (0.9, 1.5) | 0.401 | 0.7 (0.5, 1.0) | 0.053 | 1.0 (0.7, 1.3) | 0.752 | 1.0 (0.8, 1.3) | 0.896 |
| Unemployed | 0.8 (0.7, 1.2) | 0.329 | 0.7 (0.5, 0.8) | 0.001 | 0.8 (0.6, 0.9) | 0.010 | 0.9 (0.5, 1.1) | 0.253 |
| **Highest Level of Education of Household (≤ secondary school graduation as a reference group)** | | | | | | | | |
| > secondary school graduation | 1.0 (0.8, 1.2) | 0.689 | 1.5 (1.2, 1.8) | <0.001 | 1.1 (0.9, 1.3) | 0.604 | 1.0 (0.8, 1.2) | 0.811 |

**Table 4. The adjusted prevalence ratio for avoiding visiting a dental professional in the past 12 months due to cost, (2017–18, CCHS).**

| Independent variables | Adjusted Prevalence ratio (95% CI) | P-value |
|---|---|---|
| **Age (12–19 as a reference group)** | | |
| 20–39 | 2.6 (1.9, 3.5) | <0.001 |
| 40–59 | 2.1 (1.6, 2.8) | <0.001 |
| 60–79 | 1.6 (1.2, 2.1) | 0.002 |
| >80 | N/A | N/A |
| **Sex (male as a reference group)** | | |
| Female | 1.1 (1.0, 1.2) | 0.005 |
| **Household Income Group (fifth (highest) as a reference group)** | | |
| First (lowest) | 2.5 (2.1, 2.9) | <0.001 |
| Second | 2.4 (2.0, 2.8) | <0.001 |
| Third | 1.8 (1.6, 2.2) | <0.001 |
| Fourth | 1.5 (1.3, 1.8) | <0.001 |
| **Dental Insurance (yes as a reference group)** | | |
| No | 3.0 (2.7, 3.3) | <0.001 |
| **Employment status (full-time employed as a reference group)** | | |
| Part-time employed | 1.2 (1.1, 1.4) | 0.006 |
| Unemployed | 1.0 (0.9, 1.1) | 0.838 |
| **Highest level of education of household ($\leq$ secondary school graduation as a reference group)** | | |
| > secondary school graduation | 0.9 (0.8, 1.0) | 0.029 |

Among the four cycles, having no dental insurance was the strongest predictor to avoid visiting a dental professional in the last three years due to cost from 2003 to 2013–14, and as can be seen from the figure, the prevalence ratio decreased from 3.1 to 2.7. Age was the second strongest predictor to avoid visiting a dental professional in the last three years due to cost in the 2003, 2009–10 and 2013–14 cycles, whereas income was the second strongest predictor in the

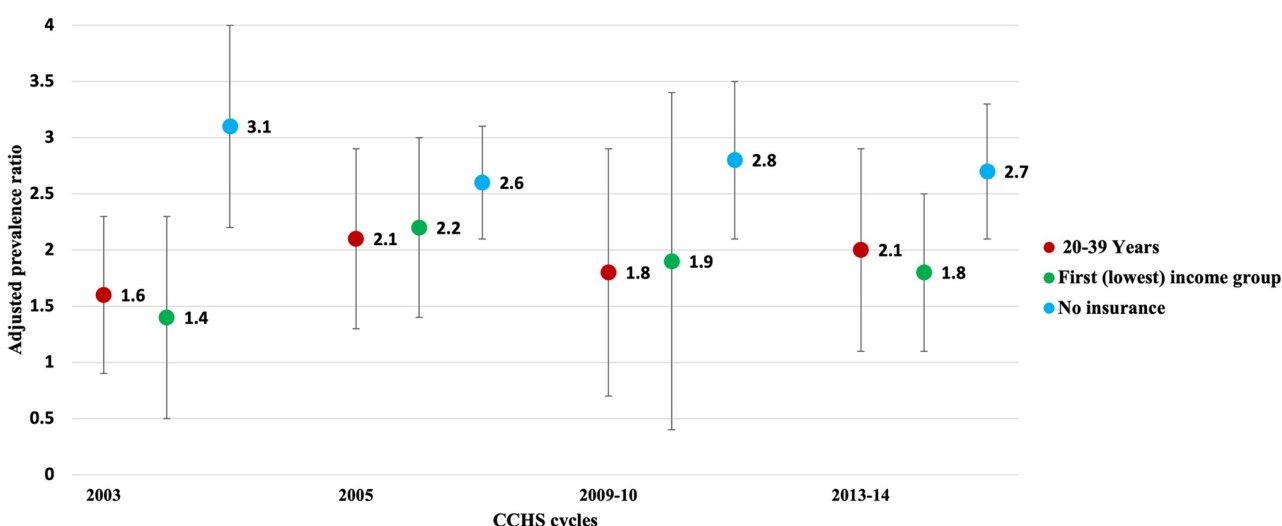

**Fig 3. The adjusted prevalence ratios for the three strongest predictors to avoid visiting a dental professional due to cost in the last three years from 2003 to 2013–14.**

2005 cycle. Over the four cycles, the prevalence ratio of individuals aged 20–39 to avoid visiting a dental professional in the last three years due to cost increased from 1.6 in 2003 to 2.1 by 2013–14. Similarly, over the given period, the prevalence ratio of those with the lowest income level to avoid visiting a dental professional in the last three years due to cost increased from 1.4 to 1.8.

## 4. Discussion

This study, analyzing data from five CCHS cycles (2003, 2005, 2009–10, 2013–14 and 2017–18), provides information regarding trends in cost barriers to dental care in Ontario. Due to methodological differences in data collection methods for the 2017–18 cycle, data for that cycle was analyzed separately from the other four cycles and was compared to the CHMS cycle 1 data from the year 2007–09. We found that the percentage of Ontarians who avoided visiting a dental professional due to cost in the three years prior to the survey raised from 22% in 2003 to 34% in 2013–14. A similar trend of increased percentage of Ontarians avoiding visit to a dental professional in the past 12 months prior to the survey was observed at 22% in 2017–18 compared to 17% for all Canadians in 2007/09 (according to the CHMS). Previous studies investigated cost barriers to dental care [4, 23]; however, to the best of our knowledge, no study has so far undertaken to explore trends in reporting financial barriers to dental care in Ontario. It is important to examine trends since they give an overview of the current situation to better understand if financial barriers to dental care are increasing, decreasing or remaining stable among Ontarians over time.

Our study identified insurance, age, and income as the three predominant predictors to experience cost barriers to dental care. Hence, this study confirms previous research emphasizing the crucial role of insurance and income to facilitate the utilization of, and access to, dental care [4, 7, 23, 25, 27]. In Canada, dental care is almost wholly privately financed, where insured individuals report less cost barriers to dental care than the uninsured [5]. Even after controlling for other factors, including age, sex, income, marital status, education and employment status, Ontarians with no dental insurance were approximately three times more likely to avoid visiting a dental professional due to cost than those with dental insurance. Over the five CCHS cycles, having no insurance was the strongest predictor to report cost barriers to dental care.

Further, this study demonstrated differences in financial barriers based on the type of insurance. While the proportion of Ontarians with employment-based insurance remained relatively the same since 2005 [10–13], there has been an overall increase in those experiencing cost barriers to dental care, suggesting a decrease in the quality of these insurance plans. Some changes, such as limiting the yearly maximum and the annual services as well as introducing co-payment or co-insurance, had negatively affected the quality of the plans [5, 27]. Moreover, there had been a gap between premiums collected and benefits paid by insurance companies over the past years [48]. Additionally, results from The SANOFI Canada Health Care Surveys highlighted a decline in employees' satisfaction and increased employers' concerns about the sustainability of their benefit plans [49, 50]. Our results also showed an increase in the proportion of Ontarians with government-based dental insurance who avoided visiting a dental professional due to cost since 2005. This increase might be due to limitations in the comprehensiveness of their insurance coverage, as often, these programs offer emergency and basic dental services. In Ontario, public dental insurance is offered to children from low-income families, individuals with disabilities, those on social assistance and recently, in 2019, to low-income seniors. The limited eligibility leaves a significant proportion of Ontarians with no dental coverage for their treatment needs [3].

Importantly, our study's findings highlight the fact that having a dental insurance though reduces financial barriers to dental care, the barriers are not addressed completely, as there are issues with the quality and level of coverage of insurance plans. With the introduction of co-insurance, co-payment and limited coverage, people have to pay more out-of-pocket for their dental care. Out-of-pocket payment for services not covered by insurance plans might pose an additional financial burden. Previous research revealed that out-of-pocket expenditures could represent a reasonable proxy of access; in other words, the more a household has to spend, the more difficult it may be to access care [51]. Such expenditure might also use up funds that could have been spent on other household services and needs. The literature suggested that over time the out-of-pocket dental expenditures have risen for households in all income quintiles; this increase was more obvious for the lower-income household [52]. Furthermore, a previous study found that the method of dental care payment influences the affordability of dental care where high-income families mostly pay for dental care through private insurance, while low-income households mostly pay out-of-pocket [7]. Furthermore, the same study found that 25% of respondents who paid out-of-pocket for their dental care reported financial barriers to dental care compared to less than one-fifth of those who paid through private insurance. Individuals who received care through public programs fell between the two groups [7].

It is important to emphasize that employment-based dental insurance is the most common form of dental insurance in Canada and is offered through one of the employee benefits that employers voluntarily offer to their employees but are not compelled to do so. The availability of employment-based insurance is related to job characteristics such as permanent, full-time, high-wage unionized jobs in large firms [15, 53, 54]. The more stable the job, the better the employee benefits. A previous study revealed that permanent employees were six times more likely to have dental benefits than temporary workers [53]. Similarly, full-time employees were four times more likely to have dental benefits [55]. Our findings showed that over the five CCHS cycles, Ontarians who worked part-time experienced more cost barriers to dental care than those who worked full-time and those who were unemployed with an overall increase in the proportion of part-time employers who report cost barriers to dental care over the survey cycles. It is interesting to note that after controlling other factors, we found that the likelihood of unemployed individuals to report financial barriers to dental care did not increase compared to full-time workers. This inconsistency might be due to the evidence that employment status itself does not impact utilization and access; however, employment status increases the odds of having dental insurance coverage [2, 15, 17].

Interestingly, in our study, Ontarians aged 20–39 years experienced more cost barriers to dental care compared to all other age groups. After controlling for other factors, Ontarians aged 20–39 years were between 1.4 to 2.6 times more likely to avoid visiting a dental professional due to cost than those 12–20 years old. In four out of five CCHS cycles, being 20–39 years old was the second predictor of experiencing financial barriers to dental care. This might be due to the fact that young adults entering the workforce often find part-time, contract work or self-employment rather than full-time permanent work with health benefits [53]. As a result, a greater proportion of this population is without employment-based insurance, which was identified as one of the prominent predictors of access to dental services.

Household income is a proxy for socioeconomic status and reflects an individual's ability to afford dental care. Consistent with findings from previous studies [4, 7, 23, 26], results from this study indicated that over the five CCHS cycles, low-income Ontarians experienced more cost barriers to dental care than other income groups. Even after controlling for other factors, low-income Ontarians were between 2 to 2.5 times more likely to avoid visiting a dental professional due to cost than high-income individuals. In four out of five CCHS cycles, low income was the third predictor to reporting cost barriers to dental care. It is essential to note

that the challenge to accessing dental care in Canada is not only related to those with low or no income but also to the working poor who neither qualify for public dental care programs nor have jobs that offer employment-based dental insurance [27, 28].

The price of dental care is an important issue when exploring the affordability of dental care in Canada, since the dental care market is predominantly private, and the payment methods are predominantly employer-sponsored insurance and out-of-pocket payment [51]. Dentists typically bill around the recommended fee guide; however, they are free to charge more according to the economics of their practice [56]. Furthermore, by tracking the consumer price index, one can find that from 1970s to the end of 20th century, prices of dental care have been less than prices of food, health care and all other products; however, since 2002, dental care prices have kept growing and have exceeded the growth in prices of all goods and services, including food and health care [57]. The Consumer Price Index (CPI) measures changes over time in the costs of a fixed basket of goods and services [58].

Similar to a previous study [23], our study demonstrates that over the five CCHS cycles, the proportion of females reporting cost barriers to dental care was slightly more than males. The difference was only significant for 2009 and 2017–18 cycles. After controlling for other factors, females were 1.2 to 1.4 times more likely to avoid visiting a dental professional due to cost than males. Despite previous research [24, 25] pointing out that females utilize dental services more often than males, our findings might reflect a higher perception of demand resulting in reporting more financial barriers to dental care.

With respect to household education, which serves as an indicator of health literacy level, one would expect that those with higher education experience less cost barriers to dental care. However, similar to previous studies [4, 7, 55], yet, contrary to [23], Ontarians with less than secondary school education show less cost barriers to dental care than those with higher than secondary school education, and the difference was statistically significant, except for the 2013–14 CCHS cycle. Further, in 2005, after controlling for other factors, Ontarians with an education higher than secondary school were 1.5 times more likely to avoid visiting a dental professional due to cost than their counterparts. However, for the other cycles both groups had an equal likelihood to report financial barriers to dental care. These findings might suggest that those with higher education have high cost-prohibitive demands and expectations for dental procedures recommended by their dentist or specialist that are not covered by their insurance plan, such as uninsured cosmetic procedures, leading them to report cost barriers to dental services. Another explanation might be related to the workforce dynamics from 2003 to 2017–18 and the availability of jobs offering employment-based dental insurance. Accordingly, a further study focusing on education and the availability of employment-based insurance is recommended. It is important to note that the question addressing household education in the CCHS asks about the highest household education. In some cases, the person with the highest level of education, for any reason, may not be working, which ultimately affects the household income and the availability of dental insurance. Additionally, skilled immigrants in Canada, who mostly have higher education from their home countries, may not be successful in securing a good enough job (with healthcare benefits; especially in their early years, as they are still struggling to navigate the system.

In terms of perceived oral health, previous studies have shown that individuals with poor oral health are less likely to receive dental care and more likely to have cost-prohibitive dental care needs [4, 7, 23, 24]. Our findings show a similar result, where over the five CCHS cycles, individuals with "fair to poor" oral health experienced more cost barriers to dental care compared to their counterparts. Unfortunately, we could not be able to include the oral health variable in the regression model, since there was no data available for 2005 and 2009–10 cycles.

Lastly, while our interpretations are not considered causal, they are hypotheses generating. There are a number of possible explanations for increasing cost barriers to dental care in Ontario. For example, changes in the labour market and the introduction of precarious employment that does not offer employment-based insurance. Another example is the changes in the quality of dental insurance where private and public plans are covering less overtime. Further, the cultural and societal drive to improve one's oral appearance might also play a role in reporting financial barriers to dental care.

### 4.1 Policy implications

This study validates that over years, that insurance and income remain the strongest predictors of reporting cost barriers to dental care in Canada. Interventions aimed at insurance and income may address the affordability of dental care and its negative implications to a certain extent. Also, if organizations responsible for large population- based surveys retain oral health and dental visits questions in surveys' common content across all jurisdictions and ensure their consistency over cycles, it can provide some robust data to assess long term inter-jurisdictional trends and act as quasi-experimental to assess the impacts of implementation of future dental public health programs.

### 4.2 Strength and limitations

Our study's strengths include a large sample size that allows us to make population-level estimations in Ontario. Further, this study contributes to the body of literature by highlighting the trends in cost barriers to dental care in Ontario over time. Lastly, in this study, since the prevalence of the outcome is high (greater than 10), we used prevalence ratios to avoid "overestimation" of the association between the independent variables and the outcome by odds ratios.

At the same time, this study also has several limitations. First, since it is a secondary data analysis of a national survey, any data entry errors made in the original survey can be neither detected nor corrected. Second, the CCHS is a cross-sectional survey; therefore, only associations can be assessed, and no causal relationship can be inferred from this study. Third, the CCHS excluded persons living on reserves and other Indigenous settlements in the provinces, full-time members of the Canadian Forces, the institutionalized population and children aged 12–17 living in foster care, which might underestimate the findings and affect the generalizability of the results to some populations. Fourth, the dependent variable and some independent variables depend on the respondent's reporting of behaviour rather than observation which might lead to measurement errors. Measurement errors could have been introduced by respondent recall errors, instability of their opinion, and the respondents' possibility of giving "socially desirable" answers. Lastly, despite reviewing and ensuring consistency of the study variables over the selected cycles, inevitable variations in the time frame for avoiding visiting a dental professional due to cost impedes comparing cycle 2017–18 with the other four cycles. However, we had an opportunity to look at and compare Ontarians who participated in the CCHS (2017–18 cycle) to Canadians in the CHMS (2007–09) since both cycles share the same question. That said, one can argue that even though the Ontarians represent the largest proportion of Canadians it may not be the most suitable comparison. Further variations in reporting of the income variable among surveys (before and after 2005) makes it harder for us to compare trends in cost barriers to dental care by household income between the two periods.

## 5. Conclusion

This study explores trends in cost barriers to dental care in Ontario by analyzing five cycles of the CCHS (2003, 2005, 2009–10, 2013–14 and 2017–18). Results indicate that self-reported

cost barriers to dental care have generally increased in Ontario but more so for those with no insurance, low income, and aged 20–39 years. These trends can provide useful information to policy makers, administrators, and dental associations and regulators, when planning future dental public health programs.

## Supporting information

**S1 Appendix. Categories of income adequacy based on total household income and the number of people in each household, CCHS, 2003.**
(DOCX)

## Author Contributions

**Conceptualization:** Mona Abdelrehim, Vahid Ravaghi, Carlos Quiñonez, Sonica Singhal.

**Data curation:** Mona Abdelrehim, Vahid Ravaghi, Carlos Quiñonez, Sonica Singhal.

**Formal analysis:** Mona Abdelrehim, Vahid Ravaghi, Carlos Quiñonez, Sonica Singhal.

**Methodology:** Mona Abdelrehim, Vahid Ravaghi, Carlos Quiñonez.

**Supervision:** Vahid Ravaghi, Carlos Quiñonez, Sonica Singhal.

**Writing – original draft:** Mona Abdelrehim, Vahid Ravaghi, Carlos Quiñonez, Sonica Singhal.

**Writing – review & editing:** Mona Abdelrehim, Vahid Ravaghi, Carlos Quiñonez, Sonica Singhal.

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
