## [Decision Letter · Decision Letter 0]

21 Feb 2023

PONE-D-22-35306Trends in self-reported cost barriers to dental care in Ontario.PLOS ONE

Dear Dr. Mona Abdelrehim,

Thank you for submitting your manuscript to PLOS ONE. After careful consideration, we feel that it has merit but does not fully meet PLOS ONE’s publication criteria as it currently stands. Therefore, we invite you to submit a revised version of the manuscript that addresses the points raised during the review process.

We look forward to receiving your revised manuscript.

Kind regards,

Hadi Ghasemi

Academic Editor

PLOS ONE

Journal Requirements:

2. Please confirm that all data sources you used were publicly available. If this is not the case, please provide information on what permissions you were granted to access these data. Furthermore, in the Methods section of your manuscript, please include the information regarding the law exempting your study from the need of ethics approval as mentioned in your ethics statement in the online submission form.

Reviewers' comments:

Reviewer's Responses to Questions

**Comments to the Author**

1. Is the manuscript technically sound, and do the data support the conclusions?

Reviewer #1: Yes

Reviewer #2: Partly

Reviewer #3: Yes

2. Has the statistical analysis been performed appropriately and rigorously? 

Reviewer #1: No

Reviewer #2: Yes

Reviewer #3: Yes

3. Have the authors made all data underlying the findings in their manuscript fully available?

Reviewer #1: Yes

Reviewer #2: Yes

Reviewer #3: Yes

4. Is the manuscript presented in an intelligible fashion and written in standard English?

Reviewer #1: Yes

Reviewer #2: Yes

Reviewer #3: Yes

5. Review Comments to the Author

Reviewer #1: Overall;

The manuscript can be very useful for policy makers in Canada, but the news value is limited for readers outside of Canada. Therefore, acceptation of this manuscript may only be useful for Canadian readers. Maybe it is better to submit this manuscript to a Canadian dental journal. Furthermore, the manuscript is quite lengthy and sometimes written more in prose style than in scientific style. Please change the style throughout the manuscript.

Introduction;

- Are dental costs not covered by insurance for underaged people in Canada (i.e. <18 years)? If so why would you include underaged people to the study?

Methods;

- How did you formulate the questions? Did you use standardized/validated questionnaires or did you just formulated questions?

- Why did you choose for a Poisson Regression? The outcome is binary. Why did you not choose a logistic regression?

- In both the methods and results section you mention p-values and significance for comparing unadjusted ratio's. How did you calculate those p-values? I cannot find the test you used for this.

- How did you check for not violating the assumptions that have to be met for a Poisson Regression?

Results;

- I do not see absolute numbers anywhere? Please report them to be able to interpret the data better.

- You made a lot of figures that could be captured in one table altogether. Please consider combining the figures in one table.

- Consider mentioning only the most important results in text and to further cite your tables instead of writing everything in text.

Discussion;

- How about residusal confounding in your results? Despite adjusting for a lot of variables, the found associations can still be explained by non-adjuted confounding variables.

Reviewer #2: 1- A good job was done. But I think the article is too long (50 pages of text in addition to 23 Figures) and boring for readers.

2- Considering that many similar articles related to the current topic have been published by the same authors (especially by Carlos Quiñonez which written bellow), it is better to remove the published results of their previous articles and only focus on the trends of cost barriers to avoiding dental treatment.

o Thompson B, Cooney P, Lawrence H, Ravaghi V, Quiñonez C. Cost as a barrier to accessing dental care: findings from a Canadian population‐based study. Journal of public health dentistry. 2014 Aug;74(3):210-8.

o Thompson B, Cooney P, Lawrence H, Ravaghi V, Quiñonez C. The potential oral health impact of cost barriers to dental care: findings from a Canadian population-based study. BMC Oral Health. 2014 Dec;14:1-0.

o Ramraj C, Sadeghi L, Lawrence HP, Dempster L, Quinonez C. Is accessing dental care becoming more difficult? Evidence from Canada's middle-income population. PloS one. 2013 Feb 20;8(2): e57377.

o Ravaghi V, Quiñonez C, Allison PJ. The magnitude of oral health inequalities in Canada: findings of the Canadian health measures survey. Community dentistry and oral epidemiology. 2013 Dec;41(6):490-8.

o Locker D, Maggirias J, Quiñonez C. Income, dental insurance coverage, and financial barriers to dental care among Canadian adults. Journal of public health dentistry. 2011 Sep;71(4):327-34.

o Sadeghi L, Manson H, Quiñonez CR. Report on access to dental care and oral health inequalities in Ontario. Public Health Ontario; 2013.

3- The objectives in the abstract section are related to previously published articles by the same authors. The purpose of the present manuscript should be written based on the title (trends and their attributable factors).

4- The Anderson model of health care utilization in the present study is the same as the Anderson model of health care utilization, which was published in their previous article:

Thompson B, Cooney P, Lawrence H, Ravaghi V, Quiñonez C. Cost as a barrier to accessing dental care: findings from a Canadian population‐based study. Journal of public health dentistry. 2014 Aug;74(3):210-8.

5- The Figures' captions don't have been numbered.

Reviewer #3: Comments to the Author

Overview comments

The methodology used is sound and logically explained, the results are reported relatively clearly and the authors provide reasonable interpretation in the discussion, offering acceptable conclusions based on the findings. However, I feel the paper needs small revisions before it will be ready for publication. Therefore, I recommend the authors consider the following points:

1) Although, the discussion has pulled all the information together in one place and the content of each paragraph is clear and it feels as if most of the arguments are well supported; but In relation to education level and cost barrier, the result showed opposite direction. It seems in this case education as a predisposing factor is somehow not properly linked to the enabling factors. Therefore, we can not only look for one reason to explain the opposite direction, rather multiple potential reasons could be the involved as a source of such finding.

The question is asking about the highest household education, how about if mother is holding the higher education and not working? What if he/she despite having a higher education cannot find a job or have a precarious employment which affects the family’s income and insurance type? Therefore, aside from the explanation provided in the second paragraph of page 35, there is a possibility for other reasons that each one could be applicable to some of the study participants. I would recommend the authors considering different possibilities and provide more explanation in the discussion section in order to strengthen this section.

2) The importance of trend analysis of national data is highly valuable in many ways. Among others, these studies can be helpful in identification and prioritization of candidate predictors, recognition of magnitude of the problem, and help with better allocation of resources to cover the unmet needs of the population. After working with multiple cycles of two large dataset (CCHS & CHMS), authors are in the best position to provide some “suggestions” for improvement of the type and structure of questions used in those surveys. These suggestions can help to generate consistent data, increase the quality, simplify the future trend investigations and provide more reliable findings.

Good luck

6. PLOS authors have the option to publish the peer review history of their article (what does this mean?). If published, this will include your full peer review and any attached files.

Reviewer #1: No

Reviewer #2: No

Reviewer #3: No

---

## [Author Response · Author response to Decision Letter 0]

28 Mar 2023

Review Comments to the Author 

Thank you for taking the time to review our manuscript. Below are our responses to your comments. 

Reviewer #1: Overall; 

The manuscript can be very useful for policy makers in Canada, but the news value is limited for readers outside of Canada. Therefore, acceptation of this manuscript may only be useful for Canadian readers. Maybe it is better to submit this manuscript to a Canadian dental journal. Furthermore, the manuscript is quite lengthy and sometimes written more in prose style than in scientific style. Please change the style throughout the manuscript. 

Thanks for your comment. We appreciate the reviewer’s point of view that this study is more relevant to Canadian audience; however, it is also applicable other OECD countries and the purpose of publishing such work outside of Canadian journal is to target the international audience too, which gives opportunities to learn from each other and to have future collaborations. In this global environment, it would be more beneficial by not being siloed. As for being more scientific in the language, authors have attempted to revise the manuscript accordingly. 

Introduction; 

- Are dental costs not covered by insurance for underaged people in Canada (i.e. <18 years)? If so why would you include underaged people to the study? 

These data are from which include children 12 years and above. In Ontario, there is a children dental program, Healthy Smiles Ontario, which covers children from low income families, with the net income of less than 27,000 for a family with two children. There are a number of children who do not qualify for this program but face affordability issues; therefore, children have been included. 

Methods; 

- How did you formulate the questions? Did you use standardized/validated questionnaires or did you just formulated questions? 

Thank you for your question. Please note that we did not formulate the questions. The Canadian Community Health survey is a national survey conducted by Statistics Canada. A list of topics was drafted by Statistics Canada and approved by an Advisory Committee consisting of representatives from health regions, all provincial and territories ministries of health and Health Canada. We have clarified it further in the manuscript. 

- Why did you choose for a Poisson Regression? The outcome is binary. Why did you not choose a logistic regression? 

Thank you for the question. Since the outcome is binary, the odds or prevalence ratios could be used to measure the association. However, our study used the prevalence ratio because it is more intuitive than the odds ratio and avoids “overestimation” of the association by odds ratio. 

Barros AJ, Hirakata VN. Alternatives for logistic regression in cross-sectional studies: an empirical comparison of models that directly estimate the prevalence ratio. BMC medical research methodology. 2003 Dec;3(1):1-3. 

Tamhane AR, Westfall AO, Burkholder GA, Cutter GR. Prevalence odds ratio versus prevalence ratio: choice comes with consequences. Statistics in medicine. 2016 Dec 30;35(30):5730-5. 

In that case, we might use the log binominal or Poisson regression. We found the same results when we calculated the unadjusted PR using the previously mentioned two methods. However, we found slight differences between the two methods when we calculated the adjusted PR. Therefore, we decided to use the Poisson regression method based on the following paper: 

Coutinho L, Scazufca M, Menezes PR. Methods for estimating prevalence ratios in cross-sectional studies. Revista de saude publica. 2008;42:992-8. 

- In both the methods and results section you mention p-values and significance for comparing unadjusted ratio's. How did you calculate those p-values? I cannot find the test you used for this. 

The P-values were calculated with the Poisson regression model. 

- How did you check for not violating the assumptions that have to be met for a Poisson Regression? 

We checked for independence and multicollinearity of the independent variables. 

Results; 

- I do not see absolute numbers anywhere? Please report them to be able to interpret the data better. 

The overall absolute numbers have now been mentioned in the beginning of the results section. Also, they could be found inside each bar in the charts. 

- You made a lot of figures that could be captured in one table altogether. Please consider combining the figures in one table. 

We addressed this comment by combining all the figures (except one) into one table. 

- Consider mentioning only the most important results in text and to further cite your tables instead of writing everything in text. 

We addressed your concern and mentioned only the most important results. 

Discussion; 

- How about residusal confounding in your results? Despite adjusting for a lot of variables, the found associations can still be explained by non-adjuted confounding variables. 

Thank you for this important comment. In our paper, we avoided including too many variables in the regression model to avoid the table two fallacy. Therefore, we selected the independent variables based on previous literature, the significance level, and multicollinearity. Hope this explanation sounds reasonable.

Reviewer #2: 

1- A good job was done. But I think the article is too long (50 pages of text in addition to 23 Figures) and boring for readers. 

Thank you for reviewing the manuscript and for your constructive feedback. The article has been shortened. Hope the edits makes it of reasonable length.

2- Considering that many similar articles related to the current topic have been published by the same authors (especially by Carlos Quiñonez which written bellow), it is better to remove the published results of their previous articles and only focus on the trends of cost barriers to avoiding dental treatment. 

o Thompson B, Cooney P, Lawrence H, Ravaghi V, Quiñonez C. Cost as a barrier to accessing dental care: findings from a Canadian population‐based study. Journal of public health dentistry. 2014 Aug;74(3):210-8. 

o Thompson B, Cooney P, Lawrence H, Ravaghi V, Quiñonez C. The potential oral health impact of cost barriers to dental care: findings from a Canadian population-based study. BMC Oral Health. 2014 Dec;14:1-0. 

o Ramraj C, Sadeghi L, Lawrence HP, Dempster L, Quinonez C. Is accessing dental care becoming more difficult? Evidence from Canada's middle-income population. PloS one. 2013 Feb 20;8(2): e57377. 

o Ravaghi V, Quiñonez C, Allison PJ. The magnitude of oral health inequalities in Canada: findings of the Canadian health measures survey. Community dentistry and oral epidemiology. 2013 Dec;41(6):490-8. 

o Locker D, Maggirias J, Quiñonez C. Income, dental insurance coverage, and financial barriers to dental care among Canadian adults. Journal of public health dentistry. 2011 Sep;71(4):327-34. 

o Sadeghi L, Manson H, Quiñonez CR. Report on access to dental care and oral health inequalities in Ontario. Public Health Ontario; 2013. 

Previous papers by Quiñonez et al. are included in the introduction or the discussion section as they have done significant work in this field in Canada. Therefore, authors believe it is challenging to remove these citations. This topic is a crucial policy topic; cost is the predominant factor limiting access to dental care in Canada as healthcare is publicly funded but dental care is not. In this regard, our paper is the first one addressing trends in cost barriers to dental care in Ontario over fifteen years. 

3- The objectives in the abstract section are related to previously published articles by the same authors. The purpose of the present manuscript should be written based on the title (trends and their attributable factors). 

Thanks for your comments, we changed the objectives as recommended. 

4- The Anderson model of health care utilization in the present study is the same as the Anderson model of health care utilization, which was published in their previous article: 

Thompson B, Cooney P, Lawrence H, Ravaghi V, Quiñonez C. Cost as a barrier to accessing dental care: findings from a Canadian population‐based study. Journal of public health dentistry. 2014 Aug;74(3):210-8. 

Anderson's model of healthcare utilization is a well-known framework used by many medical and dental researchers to conceptualize their work. 

5- The Figures' captions don't have been numbered. 

Based on reviewer #1 recommendations, we combined all the figures (except one) into one table. 

Reviewer #3: Comments to the Author 

Overview comments 

The methodology used is sound and logically explained, the results are reported relatively clearly and the authors provide reasonable interpretation in the discussion, offering acceptable conclusions based on the findings. However, I feel the paper needs small revisions before it will be ready for publication. Therefore, I recommend the authors consider the following points: 

Thank you so much for taking the time to review the manuscript and for your comments 

1) Although, the discussion has pulled all the information together in one place and the content of each paragraph is clear and it feels as if most of the arguments are well supported; but In relation to education level and cost barrier, the result showed opposite direction. It seems in this case education as a predisposing factor is somehow not properly linked to the enabling factors. Therefore, we can not only look for one reason to explain the opposite direction, rather multiple potential reasons could be the involved as a source of such finding. 

The question is asking about the highest household education, how about if mother is holding the higher education and not working? What if he/she despite having a higher education cannot find a job or have a precarious employment which affects the family’s income and insurance type? Therefore, aside from the explanation provided in the second paragraph of page 35, there is a possibility for other reasons that each one could be applicable to some of the study participants. I would recommend the authors considering different possibilities and provide more explanation in the discussion section in order to strengthen this section. 

Thank you for your comment. We added more explanation towards the end of the discussion section. 

2) The importance of trend analysis of national data is highly valuable in many ways. Among others, these studies can be helpful in identification and prioritization of candidate predictors, recognition of magnitude of the problem, and help with better allocation of resources to cover the unmet needs of the population. After working with multiple cycles of two large dataset (CCHS & CHMS), authors are in the best position to provide some “suggestions” for improvement of the type and structure of questions used in those surveys. These suggestions can help to generate consistent data, increase the quality, simplify the future trend investigations and provide more reliable findings. Good luck 

We agree on the importance of these suggestions; therefore, we provided ours in the second half of the policy implications paragraph.

---

## [Decision Letter · Decision Letter 1]

7 Jun 2023

PONE-D-22-35306R1Trends in self-reported cost barriers to dental care in Ontario.PLOS ONE

Dear Dr. Mona Abdelrehim,

Thank you for submitting your manuscript to PLOS ONE. After careful consideration, we feel that it has merit but does not fully meet PLOS ONE’s publication criteria as it currently stands. Therefore, we invite you to submit a revised version of the manuscript that addresses the points raised during the review process.

We look forward to receiving your revised manuscript.

Kind regards,

Hadi Ghasemi

Academic Editor

PLOS ONE

Journal Requirements:

Reviewers' comments:

Reviewer's Responses to Questions

**Comments to the Author**

1. If the authors have adequately addressed your comments raised in a previous round of review and you feel that this manuscript is now acceptable for publication, you may indicate that here to bypass the “Comments to the Author” section, enter your conflict of interest statement in the “Confidential to Editor” section, and submit your "Accept" recommendation.

Reviewer #4: (No Response)

Reviewer #5: All comments have been addressed

2. Is the manuscript technically sound, and do the data support the conclusions?

Reviewer #4: Yes

Reviewer #5: Yes

3. Has the statistical analysis been performed appropriately and rigorously? 

Reviewer #4: Yes

Reviewer #5: Yes

4. Have the authors made all data underlying the findings in their manuscript fully available?

Reviewer #4: Yes

Reviewer #5: Yes

5. Is the manuscript presented in an intelligible fashion and written in standard English?

Reviewer #4: Yes

Reviewer #5: Yes

6. Review Comments to the Author

Reviewer #4: This is an interesting study about Ontario's cost barriers to dental care.

The study is well-conducted, and the authors addressed most of the reviewer's comments. However, The manuscript and data are still too limited to one Province in Canada. This reviewer suggests that the authors include in the introduction additional information about the affordability and dental health published previously in other Canadian Provinces and some other countries such as Australia, the UK, etc.

Also, the discussion should include some available data from other Canadian Provinces, territories, and countries.

Reviewer #5: Very good paper. The authors addressed all comments from other reviewers of the previous version.

I agree with authors, the findings are useful to global readers, the issue of affordability of dental care is a major issue in other countries with universal health coverage that does not cover dentistry or offer limited coverage to dental care. It is also a major issue in countries with no universal coverage like USA.

I have two comments. In Tables 3 and 4, what is the point of the empty row for the reference group? either insert ‘1’ in this row (given you are using PR) or delete the row and write next to variable name (reference group XX).

There are other explanations of the lack of, or reversed education inequality in avoiding dental care that the authors may consider. The categorisation of education (high school or less vs >high school) is not the best categorisation that reflects greater opportunities for having better jobs. It is also possible that immigration status could explain the reversed, or lack of association. Immigrants could have higher education, but not necessarily successful in securing good jobs.

7. PLOS authors have the option to publish the peer review history of their article (what does this mean?). If published, this will include your full peer review and any attached files.

Reviewer #4: No

Reviewer #5: No

---

## [Author Response · Author response to Decision Letter 1]

15 Jun 2023

Thank you for taking the time to review our manuscript. The following are our comments in red.

Reviewer #4: 

This is an interesting study about Ontario's cost barriers to dental care.

The study is well-conducted, and the authors addressed most of the reviewer's comments. However, The manuscript and data are still too limited to one Province in Canada. This reviewer suggests that the authors include in the introduction additional information about the affordability and dental health published previously in other Canadian Provinces and some other countries such as Australia, the UK, etc.

Thank you for your comments. We addressed this comment by adding a paragraph about the cost barriers to dental care in the United States, Australia and the United Kingdom.

Also, the discussion should include some available data from other Canadian Provinces, territories, and countries.

We would love to include data from other provinces/territories in our discussion. However, the only available data is related to the province of Ontario or to Canada in general, which we already included in our discussion.

Reviewer #5:

Very good paper. The authors addressed all comments from other reviewers of the previous version.

I agree with authors, the findings are useful to global readers, the issue of affordability of dental care is a major issue in other countries with universal health coverage that does not cover dentistry or offer limited coverage to dental care. It is also a major issue in countries with no universal coverage like USA.

Thank you for your comments. We addressed this comment by adding a paragraph about the cost barriers to dental care in the United States, Australia and the United Kingdom.

I have two comments. In Tables 3 and 4, what is the point of the empty row for the reference group? either insert ‘1’ in this row (given you are using PR) or delete the row and write next to variable name (reference group XX).

Thank you for your comment. We changed tables 3 and 4 accordingly.

There are other explanations of the lack of, or reversed education inequality in avoiding dental care that the authors may consider. The categorisation of education (high school or less vs >high school) is not the best categorisation that reflects greater opportunities for having better jobs. It is also possible that immigration status could explain the reversed, or lack of association. Immigrants could have higher education, but not necessarily successful in securing good jobs.

Thanks for your comment. We included this explanation in our discussion.

---

## [Decision Letter · Decision Letter 2]

19 Jun 2023

Trends in self-reported cost barriers to dental care in Ontario.

PONE-D-22-35306R2

Dear Dr. Mona Abdelrehim,

We’re pleased to inform you that your manuscript has been judged scientifically suitable for publication and will be formally accepted for publication once it meets all outstanding technical requirements.

Kind regards,

Hadi Ghasemi

Academic Editor

PLOS ONE

Additional Editor Comments (optional):

Reviewers' comments:

Reviewer's Responses to Questions

**Comments to the Author**

1. If the authors have adequately addressed your comments raised in a previous round of review and you feel that this manuscript is now acceptable for publication, you may indicate that here to bypass the “Comments to the Author” section, enter your conflict of interest statement in the “Confidential to Editor” section, and submit your "Accept" recommendation.

Reviewer #4: All comments have been addressed

Reviewer #5: All comments have been addressed

2. Is the manuscript technically sound, and do the data support the conclusions?

Reviewer #4: Yes

Reviewer #5: Yes

3. Has the statistical analysis been performed appropriately and rigorously? 

Reviewer #4: Yes

Reviewer #5: Yes

4. Have the authors made all data underlying the findings in their manuscript fully available?

Reviewer #4: Yes

Reviewer #5: Yes

5. Is the manuscript presented in an intelligible fashion and written in standard English?

Reviewer #4: Yes

Reviewer #5: Yes

6. Review Comments to the Author

Reviewer #4: The authors have adequately responded to all raised questions! The manucript can be accepted for publication.

Thanks

Reviewer #5: The authors have addressed all comments and amended the manuscript accordingly. I recommend accepting the manuscript as it stands

7. PLOS authors have the option to publish the peer review history of their article (what does this mean?). If published, this will include your full peer review and any attached files.

Reviewer #4: No

Reviewer #5: No

---

## [Editor Report · Acceptance letter]

28 Jun 2023

PONE-D-22-35306R2 

Trends in self-reported cost barriers to dental care in Ontario 

Dear Dr. Abdelrehim:

I'm pleased to inform you that your manuscript has been deemed suitable for publication in PLOS ONE. Congratulations! Your manuscript is now with our production department. 

Kind regards, 

on behalf of

Dr. Hadi Ghasemi 

Academic Editor

PLOS ONE